# DISENTANGLING LEARNING REPRESENTATIONS WITH DENSITY ESTIMATION

**Eric Yeats**[1]   **Frank Liu**[2]   **Hai Li**[1]
[1]Department of Electrical and Computer Engineering, Duke University
[2]Computer Science and Mathematics Division, Oak Ridge National Laboratory
{eric.yeats, hai.li}@duke.edu   liufy@ornl.gov

## ABSTRACT

Disentangled learning representations have promising utility in many applications, but they currently suffer from serious reliability issues. We present Gaussian Channel Autoencoder (GCAE), a method which achieves reliable disentanglement via flexible density estimation of the latent space. GCAE avoids the curse of dimensionality of density estimation by disentangling subsets of its latent space with the Dual Total Correlation (DTC) metric, thereby representing its high-dimensional latent joint distribution as a collection of many low-dimensional conditional distributions. In our experiments, GCAE achieves highly competitive and reliable disentanglement scores compared with state-of-the-art baselines.

## 1 INTRODUCTION

The notion of disentangled learning representations was introduced by Bengio et al. (2013) - it is meant to be a robust approach to feature learning when trying to learn more about a distribution of data $X$ or when downstream tasks for learned features are unknown. Since then, disentangled learning representations have been proven to be extremely useful in the applications of natural language processing Jain et al. (2018), content and style separation John et al. (2018), drug discovery Polykovskiy et al. (2018); Du et al. (2020), fairness Sarhan et al. (2020), and more.

Density estimation of learned representations is an important ingredient to competitive disentanglement methods. Bengio et al. (2013) state that representations $\mathbf{z} \sim Z$ which are disentangled should maintain as much information of the input as possible while having components which are mutually *invariant* to one another. Mutual invariance motivates seeking representations of $Z$ which have *independent* components extracted from the data, necessitating some notion of $p_Z(\mathbf{z})$.

Leading unsupervised disentanglement methods, namely $\beta$-VAE Higgins et al. (2016), FactorVAE Kim & Mnih (2018), and $\beta$-TCVAE Chen et al. (2018) all learn $p_Z(\mathbf{z})$ via the same variational Bayesian framework Kingma & Welling (2013), but they approach making $p_Z(\mathbf{z})$ independent with different angles. $\beta$-VAE indirectly promotes independence in $p_Z(\mathbf{z})$ via enforcing low $D_{\mathrm{KL}}$ between the representation and a factorized Gaussian prior, $\beta$-TCVAE encourages representations to have low Total Correlation (TC) via an ELBO decomposition and importance weighted sampling technique, and FactorVAE reduces TC with help from a monolithic neural network estimate. Other well-known unsupervised methods are Annealed $\beta$-VAE Burgess et al. (2018), which imposes careful relaxation of the information bottleneck through the VAE $D_{\mathrm{KL}}$ term during training, and DIP-VAE I & II Kumar et al. (2017), which directly regularize the covariance of the learned representation. For a more in-depth description of related work, please see Appendix D.

While these VAE-based disentanglement methods have been the most successful in the field, Locatello et al. (2019) point out serious reliability issues shared by all. In particular, increasing disentanglement pressure during training doesn't tend to lead to more independent representations, there currently aren't good unsupervised indicators of disentanglement, and no method consistently dominates the others across all datasets. Locatello et al. (2019) stress the need to find the right inductive biases in order for unsupervised disentanglement to truly deliver.

We seek to make disentanglement more reliable and high-performing by incorporating new inductive biases into our proposed method, Gaussian Channel Autoencoder (GCAE). We shall explain them in

more detail in the following sections, but to summarize: GCAE avoids the challenge of representing high-dimensional $p_Z(\mathbf{z})$ via disentanglement with Dual Total Correlation (rather than TC) and the DTC criterion is augmented with a scale-dependent latent variable arbitration mechanism. This work makes the following contributions:

- Analysis of the TC and DTC metrics with regard to the curse of dimensionality which motivates use of DTC and a new feature-stabilizing arbitration mechanism
- GCAE, a new form of noisy autoencoder (AE) inspired by the Gaussian Channel problem, which permits application of flexible density estimation methods in the latent space
- Experiments[1] which demonstrate competitive performance of GCAE against leading disentanglement baselines on multiple datasets using existing metrics

## 2 BACKGROUND AND INITIAL FINDINGS

To estimate $p_Z(\mathbf{z})$, we introduce a discriminator-based method which applies the density-ratio trick and the Radon-Nikodym theorem to estimate density of samples from an unknown distribution. We shall demonstrate in this section the curse of dimensionality in density estimation and the necessity for representing $p_Z(\mathbf{z})$ as a collection of conditional distributions.

The optimal discriminator neural network introduced by Goodfellow et al. (2014a) satisfies:

$$\arg\max_{D(\cdot)} \mathbb{E}_{\mathbf{x}_r \sim X_{real}} \left[\log D(\mathbf{x}_r)\right] + \mathbb{E}_{\mathbf{x}_f \sim X_{fake}} \left[\log\left(1 - D(\mathbf{x}_f)\right)\right] \triangleq D^*(\mathbf{x}) = \frac{p_{real}(\mathbf{x})}{p_{real}(\mathbf{x}) + p_{fake}(\mathbf{x})}$$

where $D(\mathbf{x})$ is a discriminator network trained to differentiate between "real" samples $\mathbf{x}_r$ and "fake" samples $\mathbf{x}_f$. Given the optimal discriminator $D^*(\mathbf{x})$, the density-ratio trick can be applied to yield $\frac{p_{real}(\mathbf{x})}{p_{fake}(\mathbf{x})} = \frac{D^*(\mathbf{x})}{1-D^*(\mathbf{x})}$. Furthermore, the discriminator can be supplied conditioning variables to represent a ratio of conditional distributions Goodfellow et al. (2014b); Makhzani et al. (2015).

Consider the case where the "real" samples come from an *unknown* distribution $\mathbf{z} \sim Z$ and the "fake" samples come from a *known* distribution $\mathbf{u} \sim U$. Permitted that both $p_Z(\mathbf{z})$ and $p_U(\mathbf{u})$ are finite and $p_U(\mathbf{u})$ is nonzero on the sample space of $p_Z(\mathbf{z})$, the optimal discriminator can be used to retrieve the unknown density $p_Z(\mathbf{z}) = \frac{D^*(\mathbf{z})}{1-D^*(\mathbf{z})} p_U(\mathbf{z})$. In our case where $\mathbf{u}$ is a uniformly distributed variable, this "transfer" of density through the optimal discriminator can be seen as an application of the Radon-Nikodym derivative of $p_Z(\mathbf{z})$ with reference to the Lebesgue measure. Throughout the rest of this work, we employ discriminators with uniform noise and the density-ratio trick in this way to recover unknown distributions.

This technique can be employed to recover the probability density of an $m$-dimensional isotropic Gaussian distribution. While it works well in low dimensions ($m \leq 8$), the method inevitably fails as $m$ increases. Figure 1a depicts several experiments of increasing $m$ in which the KL-divergence of the true and estimated distributions are plotted with training iteration. When number of data samples is finite and the dimension $m$ exceeds a certain threshold, the probability of there being any uniform samples in the neighborhood of the Gaussian samples swiftly approaches zero, causing the density-ratio trick to fail.

This is a well-known phenomenon called the *curse of dimensionality* of density estimation. In essence, as the dimensionality of a joint distribution increases, concentrated joint data quickly become isolated within an extremely large space. The limit $m \leq 8$ is consistent with the limits of other methods such as kernel density estimation (Parzen-Rosenblatt window).

Fortunately, the same limitation does not apply to conditional distributions of many jointly distributed variables. Figure 1b depicts a similar experiment of the first in which $m-1$ variables are independent Gaussian distributed, but the last variable $\mathbf{z}_m$ follows the distribution $\mathbf{z}_m \sim \mathcal{N}(\mu = (m-1)^{-\frac{1}{2}} \sum_{i=1}^{m-1} \mathbf{z}_i, \ \sigma^2 = \frac{1}{m})$ (i.e., the last variable is Gaussian distributed with its mean as the sum of observations of the other variables). The marginal distribution of each component is

---

[1]Code available at https://github.com/ericyeats/gcae-disentanglement

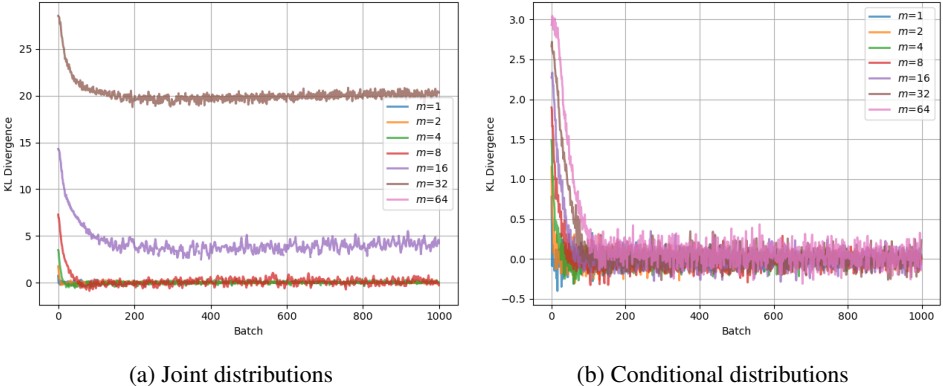

(a) Joint distributions             (b) Conditional distributions

Figure 1: Empirical KL divergence between the true and estimated distributions as training itera-
tion and distribution dimensionality increase. Training parameters are kept the same between both
experiments. We employ Monte-Carlo estimators of KL divergence, leading to transient negative
values when KL is near zero.

Gaussian, just like the previous example. While it takes more iterations to bring the KL-divergence
between the true and estimated conditional distribution to zero, it is not limited by the curse of
dimensionality. Hence, we assert that conditional distributions can capture complex relationships
between subsets of many jointly distributed variables while avoiding the curse of dimensionality.

## 3 METHODOLOGY

### ANALYSIS OF DUAL TOTAL CORRELATION

Recent works encourage disentanglement of the latent space by enhancing the Total Correlation
(TC) either indirectly Higgins et al. (2016); Kumar et al. (2017) or explicitly Kim & Mnih (2018);
Chen et al. (2018). TC is a metric of multivariate statistical independence that is non-negative and
zero if and only if all elements of $\mathbf{z}$ are independent.

$$\text{TC}(Z) = \mathbb{E}_{\mathbf{z}} \log \frac{p_Z(\mathbf{z})}{\prod_i p_{Z_i}(\mathbf{z}_i)} = \sum_i h(Z_i) - h(Z)$$

Locatello et al. (2019) evaluate many TC-based methods and conclude that minimizing their mea-
sures of TC during training often does not lead to VAE $\mu$ (used for representation) with low TC. We
note that computing $\text{TC}(Z)$ requires knowledge of the joint distribution $p_Z(\mathbf{z})$, which can be very
challenging to model in high dimensions. We hypothesize that the need for a model of $p_Z(\mathbf{z})$ is what
leads to the observed reliability issues of these TC-based methods.

Consider another metric for multivariate statistical independence, Dual Total Correlation (DTC).
Like TC, DTC is non-negative and zero if and only if all elements of $\mathbf{z}$ are independent.

$$\text{DTC}(\mathbf{z}) = \mathbb{E}_{\mathbf{z}} \log \frac{\prod_i p_{Z_i}(\mathbf{z}_i|\mathbf{z}_{\setminus i})}{p_Z(\mathbf{z})} = h(Z) - \sum_i h(Z_i|Z_{\setminus i})$$

We use $\mathbf{z}_{\setminus i}$ to denote all elements of $\mathbf{z}$ except the $i$-th element. At first glance, it appears that
$\text{DTC}(\mathbf{z})$ also requires knowledge of the joint density $p(\mathbf{z})$. However, observe an equivalent form of
DTC manipulated for the $i$-th variable:

$$\text{DTC}(Z) = h(Z) - h(Z_i|Z_{\setminus i}) - \sum_{j \neq i} h(Z_j|Z_{\setminus j}) = h(Z_{\setminus i}) - \sum_{j \neq i} h(Z_j|Z_{\setminus j}). \tag{1}$$

Here, the $i$-th variable only contributes to DTC through each set of conditioning variables $\mathbf{z}_{\backslash j}$. Hence, when computing the derivative $\partial \operatorname{DTC}(Z)/\partial \mathbf{z}_i$, no representation of $p_Z(\mathbf{z})$ is required - only the conditional entropies $h(Z_j|Z_{\backslash j})$ are necessary. Hence, we observe that the curse of dimensionality can be avoided through gradient descent on the DTC metric, making it more attractive for disentanglement than TC. However, while one only needs the conditional entropies to compute gradient for DTC, the conditional entropies alone don't measure how close $\mathbf{z}$ is to having independent elements. To overcome this, we define the summed information loss $\mathcal{L}_{\Sigma I}$:

$$\mathcal{L}_{\Sigma I}(Z) \triangleq \sum_i I(Z_i; Z_{\backslash i}) = \left[\sum_i h(Z_i) - h(Z_i|Z_{\backslash i})\right] + h(Z) - h(Z) = \operatorname{TC}(Z) + \operatorname{DTC}(Z). \quad (2)$$

If gradients of each $I(Z_i; Z_{\backslash i})$ are taken only with respect to $\mathbf{z}_{\backslash i}$, then the gradients are equal to $\frac{\partial \operatorname{DTC}(Z)}{\partial \mathbf{z}}$, avoiding use of any derivatives of estimates of $p_Z(\mathbf{z})$. Furthermore, minimizing one metric is equivalent to minimizing the other: $\operatorname{DTC}(Z) = 0 \Leftrightarrow \operatorname{TC}(Z) = 0 \Leftrightarrow \mathcal{L}_{\Sigma I}(Z) = 0$. In our experiments, we estimate $h(Z_i)$ with batch estimates $\mathbb{E}_{\mathbf{z}_{\backslash i}} p_{Z_i}(\mathbf{z}_i|\mathbf{z}_{\backslash i})$, requiring no further hyperparameters. Details on the information functional implementation are available in Appendix A.1.

EXCESS ENTROPY POWER LOSS

We found it very helpful to "stabilize" disentangled features by attaching a feature-scale dependent term to each $I(Z_i; Z_{\backslash i})$. The entropy power of a latent variable $\mathbf{z}_i$ is non-negative and grows analogously with the variance of $\mathbf{z}_i$. Hence, we define the Excess Entropy Power loss:

$$\mathcal{L}_{\text{EEP}}(Z) \triangleq \frac{1}{2\pi e} \sum_i \left[ I(Z_i; Z_{\backslash i}) \cdot e^{2h(Z_i)} \right], \quad (3)$$

which weighs each component of the $\mathcal{L}_{\Sigma I}$ loss with the marginal entropy power of each $i$-th latent variable. Partial derivatives are taken with respect to the $\mathbf{z}_{\backslash i}$ subset only, so the marginal entropy power only weighs each component. While $\nabla_\phi \mathcal{L}_{\text{EEP}} \neq \nabla_\phi \mathcal{L}_{\Sigma I}$ in most situations ($\phi$ is the set of encoder parameters), this inductive bias has been extremely helpful in consistently yielding high disentanglement scores. An ablation study with $\mathcal{L}_{\text{EEP}}$ can be found in Appendix C. The name "Excess Entropy Power" is inspired by DTC's alternative name, excess entropy.

GAUSSIAN CHANNEL AUTOENCODER

We propose Gaussian Channel Autoencoder (GCAE), composed of a coupled encoder $\phi : X \to Z_\phi$ and decoder $\psi : Z \to \hat{X}$, which extracts a representation of the data $\mathbf{x} \in \mathbb{R}^n$ in the latent space $\mathbf{z} \in \mathbb{R}^m$. We assume $m \ll n$, as is typical with autoencoder models. The output of the encoder has a bounded activation function, restricting $\mathbf{z}_\phi \in (-3, 3)^m$ in our experiments. The latent space is subjected to Gaussian noise of the form $\mathbf{z} = \mathbf{z}_\phi + \nu_\sigma$, where each $\nu_\sigma \sim \mathcal{N}(0, \sigma^2 I)$ and $\sigma$ is a controllable hyperparameter. The Gaussian noise has the effect of "smoothing" the latent space, ensuring that $p_Z(\mathbf{z})$ is continuous and finite, and it guarantees the existence of the Radon-Nikodym derivative. Our reference noise for all experiments is $\mathbf{u} \sim \operatorname{Unif}(-4, 4)$. The loss function for training GCAE is:

$$\mathcal{L}_{\text{GCAE}} = \mathbb{E}_{\mathbf{x}, \nu_\sigma} \left[ \frac{1}{n} \|\hat{\mathbf{x}} - \mathbf{x}\|_2^2 \right] + \lambda \mathcal{L}_{\text{EEP}}(Z), \quad (4)$$

where $\lambda$ is a hyperparameter to control the strength of regularization, and $\nu_\sigma$ is the Gaussian noise injected in the latent space with the scale hyperparameter $\sigma$. The two terms have the following intuitions: the mean squared error (MSE) of reconstructions ensures $\mathbf{z}$ captures information of the input while $\mathcal{L}_{\text{EEP}}$ encourages representations to be mutually independent.

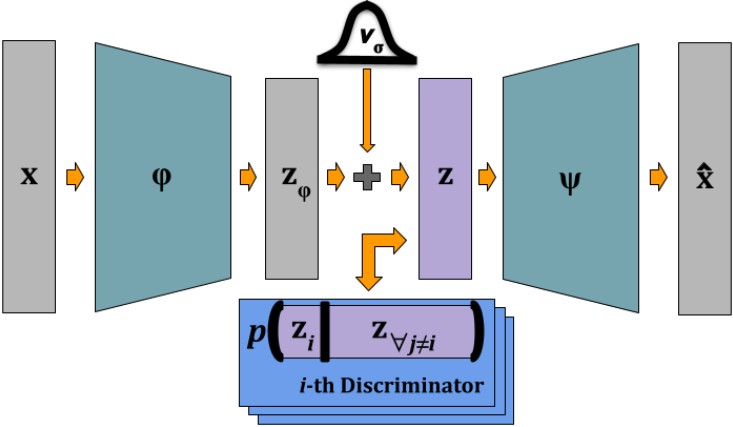

Figure 2: Depiction of the proposed method, GCAE. Gaussian noise with variance $\sigma^2$ is added to the latent space, smoothing the representations for gradient-based disentanglement with $\mathcal{L}_{\text{EEP}}$. Discriminators use the density-ratio trick to represent the conditional distributions of each latent element given observations of all other elements, capturing complex dependencies between subsets of the variables whilst avoiding the curse of dimensionality.

## 4 EXPERIMENTS

We evaluate the performance of GCAE against the leading unsupervised disentanglement baselines $\beta$-VAE Higgins et al. (2016), FactorVAE Kim & Mnih (2018), $\beta$-TCVAE Chen et al. (2018), and DIP-VAE-II Kumar et al. (2017). We measure disentanglement using four popular supervised disentanglement metrics: Mutual Information Gap (MIG) Chen et al. (2018), Factor Score Kim & Mnih (2018), DCI Disentanglement Eastwood & Williams (2018), and Separated Attribute Predictability (SAP) Kumar et al. (2017). The four metrics cover the three major types of disentanglement metrics identified by Carbonneau et al. (2020) in order to provide a complete comparison of the quantitative disentanglement capabilities of the latest methods.

We consider two datasets which cover different data modalities. The **Beamsynthesis** dataset Yeats et al. (2022) is a collection of 360 timeseries data from a linear particle accelerator beamforming simulation. The waveforms are 1000 values long and are made of two independent data generating factors: *duty cycle* (continuous) and *frequency* (categorical). The **dSprites** dataset Matthey et al. (2017) is a collection of 737280 synthetic images of simple white shapes on a black background. Each $64 \times 64$ pixel image consists of a single shape generated from the following independent factors: *shape* (categorical), *scale* (continuous), *orientation* (continuous), *x-position* (continuous), and *y-position* (continuous).

All experiments are run using the PyTorch framework Paszke et al. (2019) using 4 NVIDIA Tesla V100 GPUs, and all methods are trained with the same number of iterations. Hyperparameters such as network architecture and optimizer are held constant across all models in each experiment (with the exception of the dual latent parameters required by VAE models). Latent space dimension is fixed at $m = 10$ for all experiments, unless otherwise noted. More details are in Appendix B.

In general, increasing $\lambda$ and $\sigma$ led to lower $\mathcal{L}_{\Sigma I}$ but higher MSE at the end of training. Figure 3a depicts this relationship for Beamsynthesis and dSprites. Increasing $\sigma$ shifts final loss values towards increased independence (according to $\mathcal{L}_{\Sigma I}$) but slightly worse reconstruction error. This is consistent with the well-known Gaussian channel - as the relative noise level increases, the information capacity of a power-constrained channel decreases. The tightly grouped samples in the lower right of the plot correspond with $\lambda = 0$ and incorporating any $\lambda > 0$ leads to a decrease in $\mathcal{L}_{\Sigma I}$ and increase in MSE. As $\lambda$ is increased further the MSE increases slightly as the average $\mathcal{L}_{\Sigma I}$ decreases.

Figure 3b plots the relationship between final $\mathcal{L}_{\Sigma I}$ values with MIG evaluation scores for both Beamsynthesis and dSprites. Our experiments depict a moderate negative relationship with correlation coefficient $-0.823$. These results suggest that $\mathcal{L}_{\Sigma I}$ is a promising unsupervised indicator of successful disentanglement, which is helpful if one does not have access to the ground truth data factors.

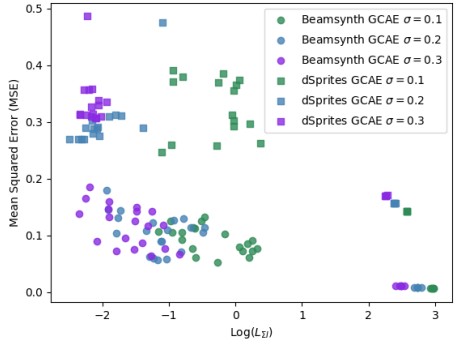 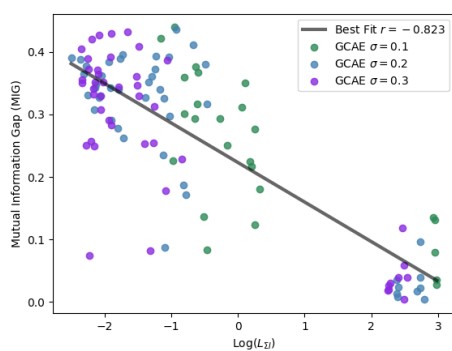

(a) Scatter plot of $\log(\mathcal{L}_{\Sigma I})$ vs. MSE for GCAE on Beamsynthesis and dSprites. Higher $\sigma$ and lower $\log(\mathcal{L}_{\Sigma I})$ (through increased disentanglement pressure) tend to increase MSE. However, the increase in MSE subsides as the model becomes disentangled.

(b) Scatter plot of $\log(\mathcal{L}_{\Sigma I})$ vs. MIG for GCAE on Beamsynthesis and dSprites (both marked with dots). There is a moderate relationship between $\log(\mathcal{L}_{\Sigma I})$ and MIG ($r = -0.823$), suggesting $\log(\mathcal{L}_{\Sigma I})$ is a promising indicator of (MIG) disentanglement.

Figure 3: Scatter plots of $\log(\mathcal{L}_{\Sigma I})$ vs MSE and MIG, respectively, as $\sigma$ is increased.

EFFECT OF $\lambda$ AND $\sigma$ ON DISENTANGLEMENT

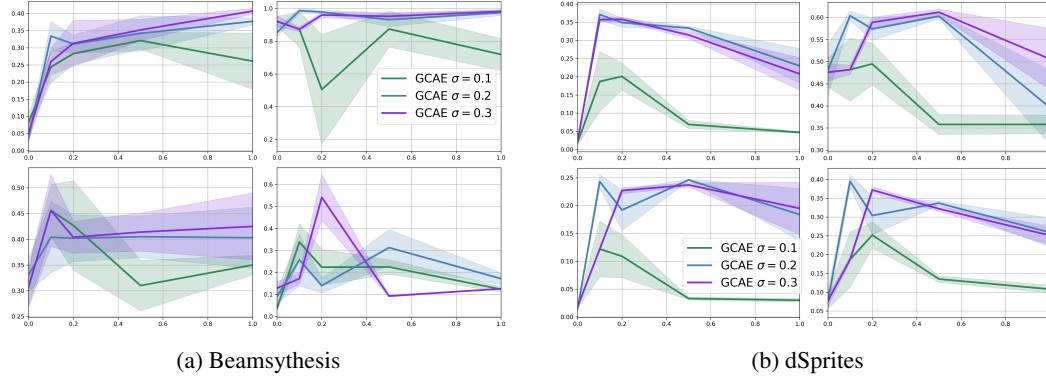

(a) Beamsythesis

(b) dSprites

Figure 4: Effect of $\lambda$ and $\sigma$ on different disentanglement metrics. $\lambda$ is varied in the $x$-axis. Starting from the top left of each subfigure and moving clockwise within each subfigure, we report MIG, FactorScore, SAP, and DCI Disentanglement. Noise levels $\sigma = \{0.2, 0.3\}$ are preferable for reliable disentanglement performance. **KEY:** Dark lines - average scores. Shaded areas - one standard deviation.

In this experiment, we plot the disentanglement scores (average and standard deviation) of GCAE as the latent space noise level $\sigma$ and disentanglement strength $\lambda$ vary on Beamsynthesis and dSprites. In each figure, each dark line plots the average disentanglement score while the shaded area fills one standard deviation of reported scores around the average.

Figure 4a depicts the disentanglement scores of GCAE on the Beamsynthesis dataset. All $\sigma$ levels exhibit relatively low scores when $\lambda$ is set to zero (with the exception of FactorScore). In this situation, the model is well-fit to the data, but the representation is highly redundant and entangled, causing the "gap" or "separatedness" (in SAP) for each factor to be low. However, whenever $\lambda > 0$ the disentanglement performance increases significantly, especially for MIG, DCI Disentanglement, and SAP with $\lambda \in [0.1, 0.2]$. There is a clear preference for higher noise levels, as $\sigma = 0.1$ generally has higher variance and lower disentanglement scores. FactorScore starts out very high on Beamsynthesis because there are just two factors of variation, making the task easy.

Figure 4b depicts the disentanglement scores of GCAE on the dSprites dataset. Similar to the previous experiment with Beamsynthesis, no disentanglement pressure leads to relatively low scores on all considered metrics ($\sim 0.03$ MIG, $\sim 0.47$ FactorScore, $\sim 0.03$ DCI, $\sim 0.08$ SAP), but introducing $\lambda > 0$ signficantly boosts performance on a range of scores $\sim 0.35$ MIG, $\sim 0.6$ FactorScore, $\sim 0.37$ SAP, and $\sim 0.45$ DCI (for $\sigma = \{0.2, 0.3\}$). Here, there is a clear preference for larger $\sigma$; $\sigma = \{0.2, 0.3\}$ reliably lead to high scores with little variance.

### COMPARISON OF GCAE WITH LEADING DISENTANGLEMENT METHODS

We incorporate experiments with leading VAE-based baselines and compare them with GCAE $\sigma = 0.2$. Each solid line represents the average disentanglement scores for each method and the shaded areas represent one standard deviation around the mean.

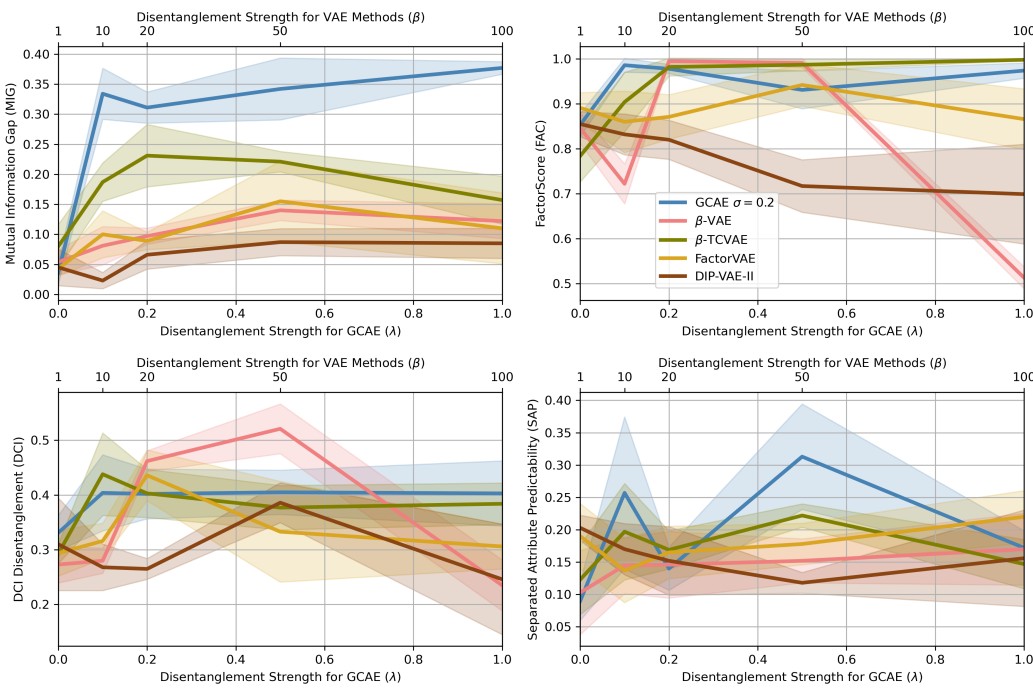

Figure 5: Disentanglement metric comparison of GCAE with VAE baselines on Beamsynthesis. GCAE $\lambda$ is plotted on the lower axis, and VAE-based method regularization strength $\beta$ is plotted on the upper axis. **KEY:** Dark lines - average scores. Shaded areas - one standard deviation.

Figure 5 depicts the distributional performance of all considered methods and metrics on Beamsynthesis. When no disentanglement pressure is applied, disentanglement scores for all methods are relatively low. When disentanglement pressure is applied ($\lambda, \beta > 0$), the scores of all methods increase. GCAE scores highest or second-highest on each metric, with low relative variance over a large range of $\lambda$. $\beta$-TCVAE consistently scores second-highest on average, with moderate variance. FactorVAE and $\beta$-VAE tend to perform relatively similarly, but the performance of $\beta$-VAE appears highly sensitive to hyperparameter selection. DIP-VAE-II performs the worst on average.

Figure 6 shows a similar experiment for dSprites. Applying disentanglement pressure significantly increases disentanglement scores, and GCAE performs very well with relatively little variance when $\lambda \in [0.1, 0.5]$. $\beta$-VAE achieves high top scores with extremely little variance but only for a very narrow range of $\beta$. $\beta$-TCVAE scores very high on average for a wide range of $\beta$ but with large variance in scores. FactorVAE consistently scores highest on FactorScore and it is competitive on SAP. DIP-VAE-II tends to underperform compared to the other methods.

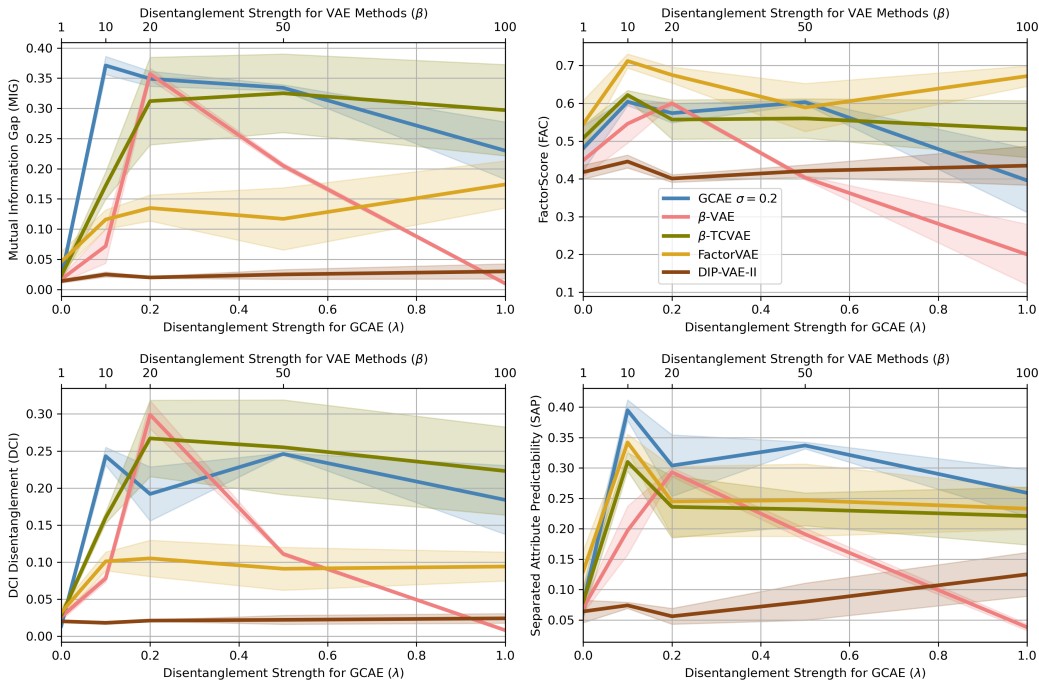

Figure 6: Disentanglement metric comparison of GCAE with VAE baselines on dSprites. GCAE $\lambda$ is plotted on the lower axis, and VAE-based method regularization strength $\beta$ is plotted on the upper axis. **KEY:** Dark lines - mean scores. Shaded areas - one standard deviation.

### DISENTANGLEMENT PERFORMANCE AS Z DIMENSIONALITY INCREASES

We report the disentanglement performance of GCAE and FactorVAE on the dSprites dataset as $m$ is increased. FactorVAE Kim & Mnih (2018) is the closest TC-based method: it uses a single mono-lithic discriminator and the density-ratio trick to explicitly approximate TC($Z$). Computing TC($Z$) requires knowledge of the joint density $p_Z(\mathbf{z})$, which is challenging to compute as $m$ increases.

Figure 7 depicts an experiment comparing GCAE and FactorVAE when $m = 20$. The results for $m = 10$ are included for comparison. The average disentanglement scores for GCAE $m = 10$ and $m = 20$ are very close, indicating that its performance is robust in $m$. This is not the case for FactorVAE - it performs worse on all metrics when $m$ increases. Interestingly, FactorVAE $m = 20$ seems to recover its performance on most metrics with higher $\beta$ than is beneficial for FactorVAE $m = 10$. Despite this, the difference suggests that FactorVAE is not robust to changes in $m$.

## 5  DISCUSSION

Overall, the results indicate that GCAE is a highly competitive disentanglement method. It achieves the highest average disentanglement scores on the Beamsynthesis and dSprites datasets, and it has relatively low variance in its scores when $\sigma = \{0.2, 0.3\}$, indicating it is reliable. The hyperparameters are highly transferable, as $\lambda \in [0.1, 0.5]$ works well on multiple datasets and metrics, and the performance does not change with $m$, contrary to the TC-based method FactorVAE. GCAE also used the same data preprocessing (mean and standard deviation normalization) across the two datasets. We also find that $\mathcal{L}_{\Sigma I}$ is a promising indicator of disentanglement performance.

While GCAE performs well, it has several limitations. In contrast to the VAE optimization process which is very robust Kingma & Welling (2013), the optimization of $m$ discriminators is sensitive to choices of learning rate and optimizer. Training $m$ discriminators requires a lot of computation, and the quality of the learned representation depends heavily on the quality of the conditional densities stored in the discriminators. Increasing the latent space noise $\sigma$ seems to make learning more

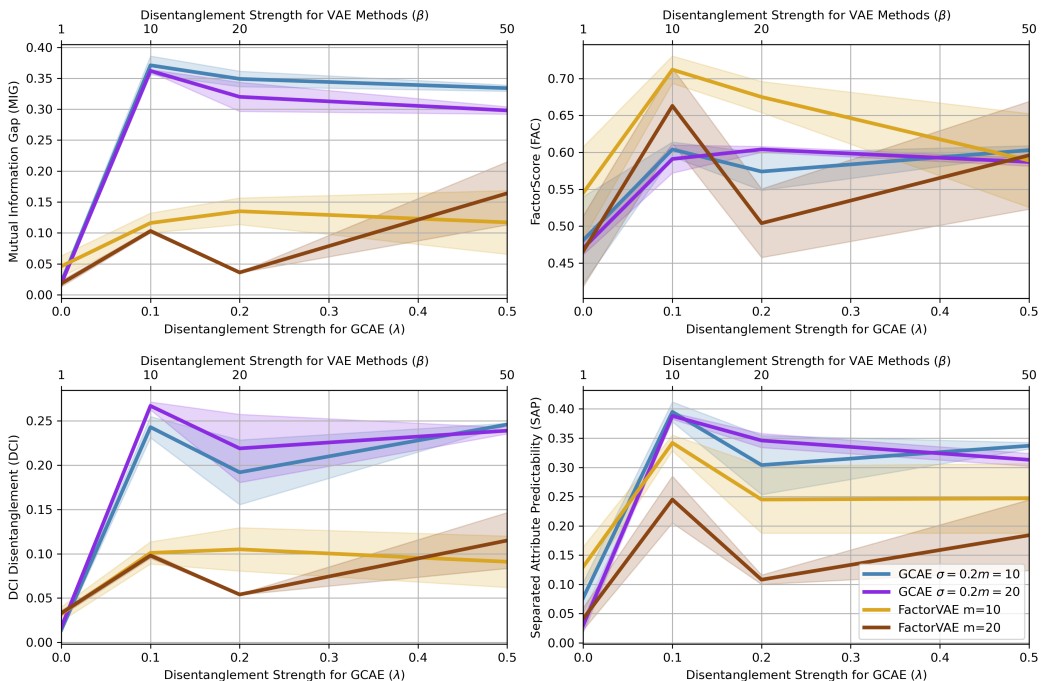

Figure 7: Comparison of GCAE with FactorVAE on dSprites as $m$ increases. $\lambda$ is plotted below, and $\beta$ is plotted above. **KEY:** Dark lines - mean scores. Shaded areas - one standard deviation.

robust and generally leads to improved disentanglement outcomes, but it limits the corresponding information capacity of the latent space.

# 6 CONCLUSION

We have presented Gaussian Channel Autoencoder (GCAE), a new disentanglement method which employs Gaussian noise and flexible density estimation in the latent space to achieve reliable, high-performing disentanglement scores. GCAE avoids the curse of dimensionality of density estimation by minimizing the Dual Total Correlation (DTC) metric with a weighted information functional to capture disentangled data generating factors. The method is shown to consistently outcompete existing SOTA baselines on many popular disentanglement metrics on Beamsynthesis and dSprites.

ACKNOWLEDGEMENTS

This research is supported by grants from U.S. Army Research W911NF2220025 and U.S. Air Force Research Lab FA8750-21-1-1015. We would like to thank Cameron Darwin for our helpful conversations regarding this work.

This research is supported, in part, by the U.S. Department of Energy, through the Office of Advanced Scientific Computing Research's "Data-Driven Decision Control for Complex Systems (DnC2S)" project.

This research used resources of the Experimental Computing Laboratory (ExCL) at ORNL. This manuscript has been authored by UT-Battelle, LLC, under contract DE-AC05-00OR22725 with the US Department of Energy (DOE). The US government retains and the publisher, by accepting the article for publication, acknowledges that the US government retains a nonexclusive, paid-up, irrevocable, worldwide license to publish or reproduce the published form of this manuscript, or allow others to do so, for US government purposes. DOE will provide public access to these results of federally sponsored research in accordance with the DOE Public Access Plan (http://energy.gov/downloads/doe-public-access-plan).

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

## A    IMPLEMENTATION

### A.1    INFORMATION FUNCTIONAL

We estimate the information between each subset of variables $I(Z_i; Z_j)$ used in $\mathcal{L}_{\Sigma I}$ with a uniform estimate of the information functional:

$$I(Z_i; Z_{\setminus i}) \approx (b - a)\mathbb{E}_{\mathbf{z}_{\setminus i} \sim \phi_\sigma(X)} \left[ \mathbb{E}_{\mathbf{u}_i \sim \text{Unif}(a,b)} \, p_{Z_i}(\mathbf{u}_i|\mathbf{z}_{\setminus i}) \left[ \log p_{Z_i}(\mathbf{u}_i|\mathbf{z}_{\setminus i}) - \log p_{Z_i}(\mathbf{u}_i) \right] \right],$$

where $(a, b)$ are the bounds of the Uniform distribution ($-4$ and $4$ in our experiments), and $p_{Z_i}(\mathbf{u}_i|\mathbf{z}_{\setminus i})$ is the conditional density of the $i$-th discriminator evaluated with noise from the Uniform distribution. 50 uniform samples are taken per batch to estimate the functional in all experiments. Furthermore, we found it beneficial (in terms of disentanglement performance) to estimate the functional using $\mathbf{z}_\phi$ (i.e., the noiseless form of $\mathbf{z}$)[2]. Gradient is only taken through the $p_{Z_i}(\mathbf{u}_i|\mathbf{z}_{\setminus i})$ term with respect to the $\mathbf{z}_{\setminus i}$ variables. The marginal entropy $h(Z_i)$ upper bounds the conditional entropy $h(Z_i|Z_{\setminus i})$ with respect to the conditioning variables, so the information functional is a natural path to maximizing $h(Z_i|Z_{\setminus i})$ and thereby minimizing DTC.

## B    MAIN EXPERIMENT DETAILS

Each method uses the same architecture (besides the $\mu$, $\log \sigma^2$ heads for the VAE) and receies the same amount of data during training. In all experiments, the GCAE AE and discriminator learning rates are $5e-5$ and $2e-4$, respectively. The VAE learning rate is $1e-4$ and the FactorVAE discriminator learning rate is $2e-4$. All methods use the Adam optimizer with $(\beta_1, \beta_2) = (0.9, 0.999)$ for the AE subset of parameters and $(\beta_1, \beta_2) = (0.5, 0.9)$ for the discriminator(s) subset of parameters (if applicable). The number of discriminator updates per AE update $k$ is set to 5 when $m = 10$ and 10 when $m = 20$. All discriminators are warmed up with 500 batches before training begins to ensure they approximate a valid density. VAE architectures are equipped with a Gaussian decoder for Beamsynthesis and a Bernoulli decoder for dSprites. SELU refers to the SeLU activation function Klambauer et al. (2017).

---

[2]Our intuition is that each $\mathbf{z}_{\setminus i}$ comes from one of the "modes" of the corresponding Gaussian-blurred distribution, ensuring that the loss is defined. This avoids the case where the learned conditional distribution is not defined when given a novel $\mathbf{z}_{\setminus i}$.

Table 1: MLP Architecture

| Dataset | GCAE Architecture | VAE Architecture |
|---|---|---|
| **Beamsynthsis** | Linear($n$, 1024), SELU | Linear($n$, 1024), SELU |
| BatchSize=64 | Linear(1024, 1024), SELU | Linear(1024, 1024), SELU |
| Mean/STD Norm | Linear(1024, 512), SELU | Linear(1024, 512), SELU |
| 2000 Iterations | Linear(512, $m$), SoftSign | $2 \times$ Linear(512, $m$) |
| | Linear($m$, 512), SELU | Linear($m$, 512), SELU |
| **dSprites** | Linear(512, 1024), SELU | Linear(512, 1024), SELU |
| BatchSize=256 | Linear(1024, 1024), SELU | Linear(1024, 1024), SELU |
| Mean/STD Norm (GCAE) | Linear(1024, $n$) | Linear(1024, $n$) |
| 20000 Iterations | | |

Table 2: Discriminator Architectures. The FactorVAE architecture follows the suggestion of Kim & Mnih (2018). The GCAE discriminator is much smaller, but there are $m$ of them compared to just 1 FactorVAE discriminator.

| GCAE Discriminator Architecture | FactorVAE Discriminator Architecture |
|---|---|
| Linear($m$, 256), SELU | Linear($m$, 1024), SELU |
| Linear(256, 256), SELU | Linear(1024, 1024), SELU |
| Linear(256, 1), Sigmoid | Linear(1024, 1024), SELU |
| - | Linear(1024, 1024), SELU |
| - | Linear(1024, 1024), SELU |
| - | Linear(1024, 1), Sigmoid |

## C  ABLATION STUDY

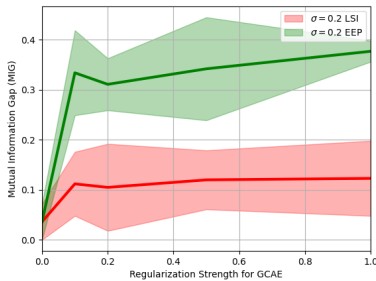

Figure 8: Ablation study: Comparison of MIG scores with and without $\mathcal{L}_{\text{EEP}}$. $\mathcal{L}_{\Sigma I}$ corresponds to direct gradient descent on $\mathcal{L}_{\Sigma I}$.

Figure 8 depicts an ablation study for training with $\mathcal{L}_{\text{EEP}}$ vs. directly with $\mathcal{L}_{\Sigma I}$. We found that training directly with $\mathcal{L}_{\Sigma I}$ promotes independence between the latent variables, but the learned variables were not stable (i.e., their variance fluctuated significantly in training). The results indicate that $\mathcal{L}_{\text{EEP}}$ is a helpful inductive bias for aligning representations with interpretable data generating factors in a way that is stable throughout training.

# D  RELATED WORK

## D.1  DISENTANGLEMENT METHODS

GCAE is an unsupervised method for disentangling learning representations - hence, the most closely related works are the state-of-the-art unsupervised VAE baselines: $\beta$-VAE Higgins et al. (2016), FactorVAE Kim & Mnih (2018), $\beta$-TCVAE Chen et al. (2018), and DIP-VAE-II Kumar et al. (2017). All methods rely on promoting some form of independence in $p_Z(\mathbf{z})$, and we shall cover them in more detail in the following sections.

### $\beta$-VAE

The disentanglement approach of $\beta$-VAE Higgins et al. (2016) is to promote independent codes in $Z$ by constraining the information capacity of $Z$. This is done with a VAE model by maximizing the expectation (on $\mathbf{x}$) of the following loss:

$$\mathcal{L}_{\beta\text{-VAE}} = \mathbb{E}_{q_\phi(\mathbf{z}|\mathbf{x})} \left[ p_\theta(\mathbf{x}|\mathbf{z}) \right] - \beta D_{\text{KL}} \left( q_\phi(\mathbf{z}|\mathbf{x}) \middle|\middle| p_\theta(\mathbf{z}) \right),$$

where $q_\phi(\mathbf{z}|\mathbf{x})$ is the approximate posterior (inferential distribution of the encoder), $p_\theta(\mathbf{x}|\mathbf{z})$ is the decoder distribution, $p_\theta(\mathbf{z})$ is the prior distribution (typically spherical Gaussian), and $\beta$ is a hyper-parameter controlling the strength of the "Information Bottleneck" Tishby et al. (2000) induced on $Z$. Higher $\beta$ are associated with improved disentanglement performance.

### FACTORVAE

The authors of FactorVAE Kim & Mnih (2018) assert that the information bottleneck of $\beta$-VAE is too restrictive, and seek to improve the reconstruction error vs. disentanglement performance tradeoff by isolating the Total Correlation (TC) component of the $D_{\text{KL}} \left( q_\phi(\mathbf{z}|\mathbf{x}) \middle|\middle| p_\theta(\mathbf{z}) \right)$ term. They employ a large discriminator neural network, the density-ratio trick, and a data shuffling strategy to estimate the TC. FactorVAE maximizes the following loss:

$$\mathcal{L}_{\text{FactorVAE}} = \mathbb{E}_{q_\phi(\mathbf{z}|\mathbf{x})} \left[ p_\theta(\mathbf{x}|\mathbf{z}) \right] - D_{\text{KL}} \left( q_\phi(\mathbf{z}|\mathbf{x}) \middle|\middle| p_\theta(\mathbf{z}) \right) - \text{TC}_\rho(Z),$$

where $\text{TC}_\rho(Z)$ is the discriminator's estimate of $\text{TC}(Z)$. The discriminator is trained to differentiate between "real" jointly distributed $\mathbf{z}$ and "fake" $\mathbf{z}$ in which all the elements have been shuffled across a batch.

### $\beta$-TCVAE

$\beta$-TCVAE Chen et al. (2018) seeks to isolate $\text{TC}(Z)$ via a batch estimate. They avoid significantly underestimating $p_Z(\mathbf{z})$, by constructing an importance-weighted estimate of $h(Z)$:

$$\mathbb{E}_{q(\mathbf{z})} \left[ \log q(\mathbf{z}) \right] \approx \frac{1}{B} \sum_{i=1}^{B} \left[ \log \frac{1}{BC} \sum_{j=1}^{B} q(\phi(x_i)|x_j) \right]$$

where $q(\mathbf{z})$ is an estimate of $p_Z(\mathbf{z})$, $B$ is the minibatch size, $C$ is the size of the dataset, $\phi(x_i)$ is a stochastic sample from the $i$-th $x$, and $q(\phi(x_i)|x_j)$ is the density of the posterior at $\phi(x_i)$ when $\mathbf{x} = x_j$.

This estimate is used to compute an estimate of $\text{TC}(Z)$, and the following loss is maximized:

$$\mathcal{L}_{\beta\text{-TCVAE}} = \mathbb{E}_{q_\phi(\mathbf{z}|\mathbf{x})} \left[ p_\theta(\mathbf{x}|\mathbf{z}) \right] - I_q(Z; X) - \beta \text{TC}_\rho(Z) - \sum_{j=1}^{m} D_{\text{KL}} \left( q(\mathbf{z}_j) \middle|\middle| p(\mathbf{z}_j) \right),$$

where $I_1(Z; X)$ is the "index-code" mutual information, $\text{TC}_\rho(Z)$ is an estimate of $\text{TC}(Z)$ computed with their estimate of $q(\mathbf{z})$, $\beta$ is a hyperparameter controlling $\text{TC}(Z)$ regularization, and $\sum_{j=1}^{m} D_{\text{KL}} \left( q(\mathbf{z}_j) \middle|\middle| p(\mathbf{z}_j) \right)$ is a dimension-wise Kullback-Leibler divergence.

DIP-VAE-II

The approach of DIP-VAE-II is that the aggregate posterior of a VAE model should be factorized in order to promote disentanglement Kumar et al. (2017). This is done efficiently using batch estimates of the covariance matrix. The loss to be maximized for DIP-VAE-II is:

$$
\mathcal{L}_{\text{DIP-VAE-II}} =
$$

$$
\mathbb{E}_{q_\phi(\mathbf{z}|\mathbf{x})}\left[p_\theta(\mathbf{x}|\mathbf{z})\right] - D_{\text{KL}}\left(q_\phi(\mathbf{z}|\mathbf{x})\big|\big|p_\theta(\mathbf{z})\right) - \beta\left(\sum_{i=1}^{m}\left[\text{Cov}(\mathbf{z}_{ii}) - 1\right]^2 + \sum_{i=1}^{m}\sum_{j\neq i}\left[\text{Cov}(\mathbf{z}_{ij})\right]^2\right).
$$

Hence, the covariance matrix of the sampled representation $\mathbf{z}$ should be equal to the identity matrix. $\beta$ is a hyperparameter controlling regularization strength. We did not consider DIP-VAE-I since it implicitly assumes knowledge of how many data generating factors there are.

## D.2 DISENTANGLEMENT METRICS

We evaluate GCAE and the leading VAE baselines with four metrics: Mutual Information Gap (MIG), FactorScore, Separated Attribute Predictability (SAP), and DCI Disentanglement.

### MUTUAL INFORMATION GAP

MIG is introduced by Chen et al. (2018) as an axis-aligned, unbiased, and general detector for disentanglement. In essence, MIG measures the average gap in information between the latent feature which is most selective for a unique data generating factor and the latent feature which is second runner up. MIG is a normalized metric on $[0, 1]$, and higher scores indicate better capturing and disentanglement of the data generating factors. MIG is defined as follows:

$$
\text{MIG}(Z, V) \triangleq \frac{1}{K}\sum_{k=1}^{K}\frac{1}{H(V_k)}\left(I(Z_a; V_k) - I(Z_b; V_k)\right),
$$

where $K$ is the number of data generating factors, $H(V_k)$ is the discrete entropy of the $k$-th data generating factor, and $\mathbf{z}_a \sim Z_a$ and $\mathbf{z}_b \sim Z_b$ (where $a \neq b$) are the latent elements which share the most and next-most information with $\mathbf{v}_k \sim V_k$, respectively.

For Beamsynthesis, we calculate MIG on the full dataset using a histogram estimate of the latent space with 50 bins (evenly spaced maximum to minimum). For dSprites, we calculate MIG using 10000 samples, and we use 20 histogram bins following Locatello et al. (2019).

### FACTOR SCORE

FactorScore is introduced by Kim & Mnih (2018). The intuition is that change in one dimension of $Z$ should result in change of at most one factor of variation. It starts off by generating many batches of data in which one factor of variation is fixed for all samples in a batch. Then the variance of each dimension on each batch is calculated and normalized by its standard deviation (without interventions). The index of the latent dimension with smallest variance and the index of the fixed factor of variation for the given batch is used as a training point for a majority-vote classifier. The score is the accuracy of the classifier on a test set of data.

For Beamsynthesis, we train the majority-vote classifier on 1000 training points and evaluate on 200 separate points. For dSprites, we train the majority-vote classifier on 5000 training points and evaluate on 1000 separate points.

### SEPARATED ATTRIBUTE PREDICTABILITY

Separated Attribute Predictability (SAP) is introduced by Kumar et al. (2017). SAP involves creating a $m \times k$ score matrix, where $ij$-th entry is the "predictability" of factor $j$ from latent element $i$.

For discrete factors, the score is the balanced classification accuracy of predicting the factor given knowledge of the $i$-th latent, and for continuous factors, the score is the $R$-squared value of the $i$-th latent in (linearly) predicting the factor. The resulting score is the difference in predictability of the most-predictive and second-most predictive latents for a given factor, averaged over all factors.

For Beamsynthesis, we use a training size of 240 and a test size of 120. For dSprites, we use a training size of 5000 and a test size of 1000.

### DCI DISENTANGLEMENT

DCI Disentanglement is introduced by Eastwood & Williams (2018). It complements other metrics introduced by the paper: completeness and informativeness. The intuition is that each latent variable should capture at most one factor. $k$ decision tree regressors are trained to predict each factor given the latent codes $\mathbf{z}$. The absolute importance weights of each decision tree regressor are extracted and inserted as columns in a $m \times k$ importance matrix. The rows of the importance matrix are normalized, and the (discrete) $k$-entropy of each row is computed. The difference of one and each row $k$-entropy is weighted by the relative importance of each row to compute the final score.

For Beamsynthesis, we use 240 training points and 120 testing points. For dSprites, we use 5000 training points and 1000 testing points.

## E  TRAINING TIME COMPARISON

Table 3: Comparison of training times of the discriminator-based disentanglement algorithms on Beamsynthesis. Latent space size is fixed to $m = 10$ and discriminator training iterations is fixed to $k = 5$.

| Method | Average (s) | Standard Deviation (s) |
|--------|-------------|------------------------|
| GCAE | 955.0 | 13.6 |
| FactorVAE | 1024.4 | 5.8 |

