# OpenReview forum: "Disentangling Learning Representations with Density Estimation"
_ICLR.cc/2023/Conference — ICLR 2023 poster_

### Official Review · Reviewer_qWQn · 2022-10-17

**Confidence:** 4
**Correctness:** 3
**Technical Novelty And Significance:** 3
**Empirical Novelty And Significance:** 4
**Recommendation:** 6

**Clarity, Quality, Novelty And Reproducibility:**

The clarity and quality of the manuscript could be improved (see weaknesses above). To the best of my knowledge, the method is novel, albeit it could be related better to prior work. Reproducibility is good: authors provide the details of their training procedure.

**Strength And Weaknesses:**

Strengths:
- The observation that TC depends on the quality of the full density estimate is a keen one, and the solution based on combining TC with DTC is non-trivial and novel.
- The proposed method is introduced in great detail. The mathematical notation is clear and consistent.
- Experiments use strong baselines: $\beta$-VAE and its variants. The proposed method consistently outperforms these baselines.
- Empirical studies of the role of hyperparameters $\sigma$ and $\lambda$.
- A good critical discussion of the results, as well as the limitations of the method.

Weaknesses:
- Related work limited to a single paragraph in the introduction. I would've liked to see a broader discussion of disentanglement methods, representation learning methods, and density estimation methods. For density estimation, especially methods that model the density as a sequence of conditional distributions (autoregressive density estimators).
- The KL divergence values are sometimes _negative_ in Figure 1(b): how is this possible?
- Several choices in the paper are not clearly motivated, including the density ratio estimation as the density estimation method, the loss weighting scheme in equation 3, and the autoencoder with Gaussian noise as the representation learning method.
- Density ratio estimation is characterized several times as "non-parametric", even though we're parametrizing the density (albeit implicitly) via the discriminator neural net.
- I am not convinced we're comparing apples to apples in Figure 1: in (a) we're estimating an $m$-dimensional density, while in (b) we're estimating a 1-dimensional density. Estimating the $m$-dimensional density as a series of conditional densities in (b) would be more convincing.
- Authors do discuss that the method is computationally expensive (need to train $m$ discriminators), but there are no runtime metrics reported in the experiments section.

Questions:
- Have you tried using GCAE as a generative model (as you suggest in the discussion)?
- Do you have intuition for why in Figure 4, (b) part especially, the MIG is going down for $\alpha > 0.2$? Is it because we're "over-disentangling", and the model struggles to capture any info about the true factors?

Minor:
- Citation style: consider making use of `\citep` and `\citet`.
- Define $m$ in the caption of Figure 1.
- A figure on page 6 has no caption. It is also not clear what the different dots that have the same color represent.
- Define $h$ in the first equation of section 3.

**Summary Of The Paper:**

The paper considers the problem of disentangling representations, where we aim to learn representations that have independent features. Authors start with the common learning objective that encourages disentanglement, the Total Correlation (TC), and hypothesize that an important disadvantage of this objective is that it requires an estimate of the latent density. This density is difficult to compute for high-dimensional representations due to the curse of dimensionality. Authors propose an alternative objective, Excess Entropy Power (EEP), that combines TC with Dual Total Correlation (DTC), and can be computed using only the estimates of _conditional_ densities in the latent space. Such conditional densities are often easier to estimate, as also shown by authors in a toy experiment. Authors run experiments on two datasets with known latent factors, observing that EEP improves upon strong VAE-based baselines.

**Summary Of The Review:**

The proposed method solves an important problem, is novel, and is evaluated well. However, several choices in the paper are not very well motivated, and the clarity and quality of the manuscript could be improved. As it stands, I consider the paper to be below the bar for acceptance, but would be willing to reconsider if the authors addressed the points raised above.

---

Authors have responded to some of my points, especially in regards to motivating the choices in the paper, and I have increased my score as a result. I believe the paper can be made stronger still by better connecting to methods in density estimation and studying the pros and cons of autoregressive density estimation in this context, but I'd now support it getting accepted.

---

> ### Author Response · Authors · 2022-11-16
> **Addressing the Weaknesses Section**
>
> Thank you for pointing this out – we will add a more in-depth discussion of the related work to the paper, and we will focus on the unsupervised VAE variants and autoregressive methods. The method differs from autoregressive approaches in that density is measured in the latent space rather than the data, and the strategy is to make each latent variable independent of all other latent variables. Hence, the joint density of z in a disentangled GCAE is computed as the product of the marginals. This contrasts the approach of autoregressive density estimators, which sequentially construct a joint density of (highly correlated) x using the chain rule of probability.
>
>
> You are correct that KL divergence is non-negative. We report negative values in some of these experiments because we employ Monte Carlo estimates of the entropy and cross-entropy of the data under the true distribution and estimated distributions, respectively. These estimates are then used to compute the KL. Fluctuations in each estimate can cause the reported value to be slightly negative when KL is near zero.
>
>
> Thank you for pointing this out. We chose to measure density using the density-ratio trick for several reasons. 1) The density-ratio trick is used in FactorVAE, providing a more direct comparison with similar methods. 2) It requires fewer assumptions & hyperparameters compared to other density estimation methods such as Gaussian Mixture Models, VAE, Normalizing Flow, etc. 3) It avoids a prior assumption. 4) We demonstrated that it captures low-dimensional densities (with arbitrary numbers of conditioning variables) very well – this is all that is needed for the disentanglement task.
>
>
> The intuition for equation (3) is that high-variance “learned” latents (i.e., not “dead” neurons) will exhibit stronger DTC (disentanglement) pressure on similar latents. Conversely, low-variance (“dead”) latents will not disrupt a high-variance latent that captures similar concepts. This was loosely inspired by the arbitration mechanisms for cellular differentiation in biology. We found that this asymmetric weighting in the EEP loss significantly boosts the disentanglement performance on both datasets.
>
>
> You make a good point. We use an autoencoder (AE) with Gaussian noise to provide the most flexibility in terms of learned latent density. The use of AE allows for any latent density to be learned (in contrast to VAE, which conforms the latent density with a Gaussian prior). This complements the extreme flexibility of the discriminators for density estimation. Gaussian noise is required to ensure the density of the latent space is smooth and finite. In the case that latent variables are discrete-valued or “dead”, a noiseless AE would have degenerate densities (and the corresponding entropies would be negative infinity).
>
>
> Density estimation with discriminators is non-parametric in the sense that the discriminators do not parameterize a certain family or mixture of densities (in terms of mean, variance, alpha, beta, etc.). While the discriminators have finite capacity, we argue that it is fair to call the densities captured by the discriminators non-parametric since MLPs are universal function approximators.
>
>
> You are right that we are not comparing apples to apples in Figure 1. The intended message in Figure 1 is that conditional distributions can be used to capture complex relationships among many variables while avoiding the curse of dimensionality. The drawback is that multiple conditional distributions are needed to capture all relationships between the variables. In our case, we focus on m 1-dimensional conditional distributions to perform disentanglement, but the method could be extended to conditional distributions of arbitrary dimension (this would correspond to disentangling vector chunks of the latent space). We expect that the conditional distributions method would fail if the distribution we are trying to estimate given the conditioning variables is too high-dimensional (just like the full joint distribution case). We will clarify this in the paper.
>
>
> We will include training time in the supplementary material – please stay tuned as we collect more data before the deadline.

---

> > ### Author Response · Authors · 2022-11-16
> > **Addressing Questions**
> >
> > A generative model version of GCAE is outside the scope of the paper, and we are investigating that for future work. However, the idea is that GCAE could be transformed into a generative model by doing the following: (1) replace the decoder with one which represents p(x|z) (like in VAE), (2) once Z is independent, compute p(z) from the discriminators, and (3) use the Gaussian distribution induced by the noise as the approximate posterior q(z|x). One can then derive a lower bound for the data log-likelihood.
> >
> >
> > Yes, our intuition on the trend in Figure 4b is that as disentanglement strength increases too far, it becomes more difficult for the latent space to capture all of the information of X. This can be explained by the decomposition of our training criterion into Reconstruction(X) + Disentanglement(Z). Since the MSE(X, X’) upper bounds the conditional entropy power 1/(2 pi e) e^(2h(X|Z)), minimizing MSE(X, X’) corresponds to maximizing the information I(X; Z) = h(X) – h(X|Z). If we place more weight on disentanglement, the fixed point of the solution will be biased less in favor of I(X;Z), leading to decreased MIG scores.

---

> > ### Comment · Reviewer_qWQn · 2022-12-12
> > **Re: Addressing the Weaknesses Section**
> >
> > I thank the authors for their response. Some of my points have been addressed, and I'll increase my score. A few remaining comments:
> >
> > > The method differs from autoregressive approaches in that density is measured in the latent space rather than the data
> >
> > Not sure I follow: autoregressive approaches can estimate any density you give them, be it the data density or the latent density. I suppose the key difference is that you're not interested in the density _per se_, you just want to regularize your latent space to have few correlations between variables, as estimated in an auto-regressive fashion.
> >
> > > The use of AE allows for any latent density to be learned
> >
> > I am not sure this is trivial to see/show. Are there any proofs of this you can point to?
> >
> > > in contrast to VAE, which conforms the latent density with a Gaussian prior
> >
> > This doesn't mean a VAE can't learn a complex posterior if favored by the data. Besides, VAEs with complex (e.g. flow-based) priors/posteriors have been studied.
> >
> > > While the discriminators have finite capacity, we argue that it is fair to call the densities captured by the discriminators non-parametric since MLPs are universal function approximators.
> >
> > This is _not_ what "non-parametric" usually implies, and I'd encourage the authors to fix this in the manuscript to avoid confusing the reader. The number of effective "parameters" in a non-parametric model must grow with sample size, like in KDE. This is most certainly _not_ the case for any finite MLP (and the density it parametrizes), and these models are never referred to as non-parametric in literature.
> >
> > > The intended message in Figure 1 is that conditional distributions can be used to capture complex relationships among many variables while avoiding the curse of dimensionality.
> >
> > I don't think Figure 1 is achieving this, though. What it _is_ showing is that estimating a 1-dimensional density is easier than estimating an $m$-dimensional density, which isn't surprising. If you compared estimating an $m$-dimensional density _directly_ with e.g. an $m$-dimensional MoG vs. estimating the same $m$-dimensional density with a sequence of conditional 1-dimensional MoGs, maybe keeping the total number of components fixed for both methods, this would be more convincing. Besides, estimating a high-dimensional density as a sequence of 1-dimensional conditional densities is common in density estimation literature (autoregressive models), and if you connected your work to this line of research this figure might not even be needed.

---

> > > ### Author Response · Authors · 2022-12-12
> > > **Author response to 2nd round discussion (qWQn)**
> > >
> > > Thank you very much for taking the time to check the revision and to give additional valuable feedback on this work.
> > >
> > >
> > > **Distinction with autoregressive models**
> > >
> > > You’re correct that Autoregressive approaches can be used to estimate any density; what we meant in this distinction is in how they are typically used (i.e., not to measure $p(z)$). A further distinction with autoregressive approaches is in how we partition the joint density. Our work estimates the product of (complete) conditionals as is done in DTC, rather than a sequence of conditionals as is done in autoregressive models. The way we use conditional densities does not impose an ordering of the joint variables, unlike in autoregressive models. Furthermore, our method has clear utility for disentanglement via the connection with DTC, whereas the application of autoregressive techniques for disentanglement is unclear.
> > >
> > > **Choice of AE base over VAE base**
> > >
> > > The choice of AE rather than VAE means that we do not put an explicit constraint on the form of latent density learned by the model. It is commonly agreed in unsupervised disentanglement literature that disentangled representations should be factorized. However, for the purpose of disentanglement there is no need for the components of the factorized distribution to follow a prescribed structure, so our choice of AE here relaxes this constraint. We acknowledge that GCAE implicitly constrains possible latent densities by restricting latent encodings within [-3, 3] and employing fixed Gaussian noise in Z, but we argue that these constraints are much milder than those imposed by VAE.
> > >
> > > **VAE with flexible prior**
> > >
> > > VAEs with flow-based priors/posteriors have been studied [Huang et al 2017, “Learnable Explicit Density for Continuous Latent Space and Variational Inference”], but (to our knowledge) this is the first time that density estimation of the latent space is used explicitly for disentanglement. Our application of density estimation as the product of conditionals has clear application for disentanglement, whereas it is not clear how a flow would be used in our problem setting. We leave a more VAE-like version of GCAE (which forms a generative model for X) to future work.
> > >
> > > **This is not what "non-parametric" usually implies**
> > >
> > > We see your point on how effective “parameters” of non-parametric methods should grow with sample size, and we will change the wording in the manuscript.
> > >
> > > **Figure 1**
> > >
> > > What Figure 1 shows is that our discriminator-based approach can capture the $p(z_i|z_j\forall j \neq i)$ relationship very well (in terms of KL-divergence), and that its optimal performance with a fixed discriminator capacity seems reasonably independent of the number of conditioning variables. We agree that the figure is not particularly surprising considering non-parametric density and autoregressive model literature, but we believe it sets the stage well for the rest of the paper (conditional densities with discriminators are scalable and they are all you need for minibatch stochastic gradient descent of DTC). We can mention the connection with autoregressive approaches in a final version.

---

### Official Review · Reviewer_cTbH · 2022-10-24

**Confidence:** 3
**Correctness:** 3
**Technical Novelty And Significance:** 3
**Empirical Novelty And Significance:** 3
**Recommendation:** 5

**Clarity, Quality, Novelty And Reproducibility:**

Overall, the paper’s idea is interesting, while descriptions can be less clear at places

— In the bottom of page 4, the authors wrote ‘to extract a compressed, noise-resistant representation’. Can you elaborate why the representation is noise-resistant?

— Figure two and page 4 descriptions are not aligned. For example, in the figure 2, the encoder with parameter phi (lower case) would generate representation z_phi; In the main text description, you are using uppercase phi. More importantly, u is missing in Figure 2, and the figure 2 is not so informative.

—It’s unclear to me if m<<n (in page 4) is always realistic.

--Figure 3 caption is missing.


**Strength And Weaknesses:**

The strengths include but not limited to:

-- The proposed methods are well-motivated. The authors performed preliminary experiments to show the effect of curse of dimensionality for non-parametric density estimation, which is illustrated in Figure one. Based on this fact, the authors analyze the TC and DTC, and show the merits of DTC.

-- The proposed method does achieve higher MIG compared with a few leading VAE-based approaches.


The weaknesses include:

-- There are a lot of quantitative and qualitative ways for evaluating disentangled representations. For quantitative evaluation, there are a few widely used metrics besides MIG, including explicitness (Ridgeway & Mozer, 2018), beta score (Higgins et al., 2017), SAP (Kumar et al., 2018) and DCI disentanglement (Eastwood & Williams 2018).  Typically, previous works (on VAE-based disentanglement) would try to show their methods are consistently better using multiple metrics.

-- Besides the three leading approaches (by 2018) mentioned by the authors, there are also some important works missed by the authors. For example, DIP-VAE (Kumar et al., 2017) and CCI-VAE (Burgress et al., 2017). More importantly, latest works on VAE-based disentangled representation learning has already realized the importance of using inductive biases. A recent work published in ICML (Mita et al., 2021) employs a particular form of factorized prior (also conditionally depends on auxiliary variables) towards learning an identifiable model with theoretical guarantees on disentanglement. Te authors could try to compare with some more (especially recent ones) important works using more datasets.

-- The preliminary experiments do show that higher m may cause difficulty in training/optimization for other works (especially Factor VAE), but the whole set of experiments in the paper are done under a single dimension m=10. It would be interesting to conduct more ablation studies to show the benefits of the proposed methods in terms of curse of dimensionality compared with FactorVAE and Beta VAE.

-- Some comments on ablation studies: In Figure 3, Figure 4 and Figure 5, the best performance is achieved using sigma=0.3 (at least for Beamsythesis), did you further try sigma larger than 0.3? Also, in page 5, you mentioned the importance of adding Gaussian noise – it ensures that p(z) is continuous and finite with respect to uniform density. Did you try the scenario where sigma=0?


**Summary Of The Paper:**

The authors propose learning disentangled representation via regularizing Gaussian Channel Autoencoder (GCAE) with Excess Entropy Power (EEP) loss. The authors claim their proposed method can achieve more reliable and high-performing disentanglement.
Their methods are largely motivated by the curse of dimensionality difficulty issue of the FactorVAE, one of the leading VAE-based disentangling approaches.  FactorVAE modifies beta-VAE objective by specifically penalizing the dependencies between the latent dimensions towards benefiting the Total Correlation (TC). The authors (of this paper) propose to use Dual Total Correlation (DTC) towards avoiding the curse of dimensionality.

The authors examined their proposed approaches on two widely used datasets using one metric – Mutual Information Gap (MIG). Experiments show that their approaches achieve (in average) higher MIG compared with other three leading VAE-based disentanglement approaches.


**Summary Of The Review:**

Overall, the method is novel and demonstrates its effectiveness on learning disentangled representations. However, I do have some concerns/comments as mentioned in my review. I’m slightly not inclined at this moment, but am willing to discuss with authors and change my views accordingly.

---

> ### Author Response · Authors · 2022-11-19
> **Author response to questions (cTbH)**
>
> This is a good point; we added more metrics (MIG, FactorScore, SAP, and DCI Disentanglement) in the latest revision. We also added DIP-VAE-II as a baseline. We did not add DIP-VAE-I as a baseline since it implicitly assumes knowledge of $k$, the number of data generating factors. Besides, DIP-VAE-I did not perform as well as betaVAE, FactorVAE, and betaTCVAE in the comprehensive Locatello et al (2019) paper.
>
> Mita et al 2021 (and similar inductive biases introduced by Locatello 2020) is interesting, but it is outside the scope of this paper as we only consider strictly unsupervised methods.
>
> We included an experiment with GCAE and FactorVAE for $m=20$ on dSprites in the revision. Thank you for pointing this out.
>
> We observed that $\sigma=0.4$ made it difficult for the model to learn, especially early on. Early experiments with $\sigma=0$ led to very poor disentanglement outcomes.
>
> **Later Edit for the Clarity, Novelty, and Reproducibility Section:**
>
> We apologize for missing these in the original reply, and we appreciate your eye for detail here.
>
> **Compressed & Noise Resistant**: By compressed, we are referring to latent codes typically having a much lower dimension than that of X. We consider the GCAE representations noise resistant in the Gaussian Channel sense - the latent space is trained to minimize reconstruction error of X while the latent space is subjected to Gaussian noise.
>
> **Figures 2 & 3**: Thanks for pointing out the notation inconsistencies for Figure 2 and the missing main caption for Figure 3. We apologize for these errors and we will fix them in a final version.
>
> **m << n**: In disentanglement (and AE/VAE) literature, it is typically assumed that the number of data generating factors (and hence the choice for m) is much smaller than the dimension of x [Bengio et al 2013 "Representation Learning: A Review and New Perspectives"]

---

### Official Review · Reviewer_atH7 · 2022-10-24

**Confidence:** 3
**Correctness:** 3
**Technical Novelty And Significance:** 2
**Empirical Novelty And Significance:** 2
**Recommendation:** 6

**Clarity, Quality, Novelty And Reproducibility:**

Clarity: Not very clear. The paper has confusing notations. The proposed method is not well explained.

Quality: Some claims are not supported. Experiments are not convincing.

Novelty: The paper contributes some new ideas.

Reproducibility: Good. Key details of the experiments are provided.

**Strength And Weaknesses:**

Strength:

+ It is interesting to use the dual total correlation instead of total correlation as the metric for disentangle representation learning. The metric circumvents the curse of dimensionality of estimating the joint distribution of latent representations.

+ The paper is well organized and easy to follow. The motivation is clear.

Weakness

- The paper claims the method can achieve reliable and high-performing disentanglement outcomes. It seems that only the "reliable" part is supported by higher disentanglement scores from the experiment. What is the definition of "high-performing" and how it is supported by the paper?

- The experimental results are not convincing. As stated in the paper that "there currently aren’t good unsupervised indicators of disentanglement", why only the Mutual Information Gap (MIG) is used as the disentanglement score? How about Z-Diff score [1], SAP score [2], and Factor score [3]?

[1] beta-vae: Learning basic visual concepts with a constrained variational framework. In ICLR, 2017

[2]  Variational inference of disentangled latent concepts from unlabeled observations. In ICLR, 2018

[3]  Disentangling by factorising. In ICML, 2018.

- Some notations are confusing. For example, $DTC(z) = DTC_i(z)$ in Eq. (1). What do $z_{\forall j\neq i}$ and $z_{k\neq j}$ mean and what are their difference? I don't follow  the discussion from $I(z_i; z_{\forall j\neq I})$ in Eq.(2) to $I(z_i;z_j)$.

- The motivation for the major loss term in Eq. (3) is not clear. What is the definition of feature-scale dependent term and why it is helpful?


**Summary Of The Paper:**

To address the reliability issue of learning independent representations, the paper proposes to regularize a noisy autoencoder with a new regularization term based on dual total correlation. This results in a new Gaussian Channel Autoencoder, which does not need to estimate the joint distribution of the latent representation and achieves higher disentanglement scores.

**Summary Of The Review:**

The paper contributes some new ideas to disentangled representation learning. However, some claims are not well supported and the experiments are weak. The writing could also be improved by clearly defining the notations and clarifying some confusing terms.

---

> ### Author Response · Authors · 2022-11-19
> **Author response to questions (atH7)**
>
> $\text{DTC}_i(z)$ is equivalent to $\text{DTC}(z)$, but it is re-written to emphasize that the only terms that depend on the $z_i$ are the conditional entropies $h(z_j|z_k \forall k \neq j)$. Here, $z_i$ is present in the conditioning variables. This has the important implication that gradient for DTC does not depend on the joint density.
>
> We added additional motivation / intuition for the EEP loss in the latest revision - please take a look near its definition.
>
> To us, reliability is characterized by having low variance in scores, while high-performing is having high average scores. GCAE has high average scores with relatively low variance (on MIG, FactorScore, SAP, and DCI Disentanglement in the latest revision), so we consider it to be high-performing and reliable compared to the other methods.

---

> > ### Comment · Reviewer_atH7 · 2022-11-22
> > **Response to Authors**
> >
> > Thank you for the clarifications and making appropriate changes to the the paper. Most of my concerns have been addressed.
> > Overall, I have raised my rating.

---

### Official Review · Reviewer_d23T · 2022-10-25

**Confidence:** 5
**Correctness:** 4
**Technical Novelty And Significance:** 2
**Empirical Novelty And Significance:** 2
**Recommendation:** 6

**Clarity, Quality, Novelty And Reproducibility:**

The work needs more finetuning and structuring.
Example:
1) Page 6: "The tightly grouped samples in the lower right of the plot correspond with lambda = 0, and incorporating any lambda > 0 leads to a decrease in L_sumI and increase in MSE. As lambda is increased further the MSE increases only slightly as the average L_sumI decreases significantly." This line is unclear in context with lambda and figure 3a.
2) Page6: "Figure 3b plots the relationship between end-of-training L_ΣI values with MIG evaluation scores for both Beamsynthesis and dSprites". But in that figure square box representing dsprites as shown in fig 3a is missing.
3) In eq1, the authors have mentioned DTC(z) = DTC_i(z); I think a few lines describing why that is necessary for the readers.


**Strength And Weaknesses:**

Strength

1) The analysis is shown in section 3, which derives the DTC and its comparison against total correlation. DTC in eqn 1. removes the requirement to compute p(z). DTC requires computing the conditional entropies for disentanglement.
2) Propose a new "Gaussian Channel Autoencoder" (GCAE). As the name suggests, gaussian noise with variance sigma^2 is added to smooth the representations. And in Sec 2, the authors have detailed how they use the discriminator-based method, which applies the density-ratio trick and the Radon-Nikodym theorem to estimate the density of samples from an unknown distribution.
3) The authors conduct experiments on Beamsynthesis (time series dataset) and dsprites (synthetic images) to show GCAE is better than previous baselines. Fig 3a and 3b show the scatter plot of GCAE for summed information loss against mean squared error (MSE) and mutual information gap (MIG). GCAE performs better than baseline models for Beamsynthesis, while in the case of dsprites it does better only for a small range of lambda (hyperparameter of summed information loss).



Weakness

1) The novelty of the paper is limited. The authors claim that using DTC  removes the constraint of estimating p(z), which is difficult compared to the conditional distribution. However, computing "m" conditional distributions needs "m" discriminators, which are sensitive to hyperparameters. It also requires a lot of data and time.
2) The authors use only the MIG metric to show disentanglement. The authors should have compared against other metrics such as DCI, SAP, etc. Please refer to "Measuring Disentanglement: A Review of Metrics" to understand why there is not one disentangling metric and each metric explores different aspects.
3) The datasets used for the experiments are limited. The results work well for Beamsynthesis, not for dsprites. It can be seen in figure 4a and 4b. The scatter plot in figure 3b for beam synthesis shows a negative correlation coefficient of -0.823. There is no mention of dsprites.
4) No qualitative results were shown for dsprites and no results on the latent traversal on these z axes. Since the authors chose m=10 as the latent dimension, and dsprites has five factors of variation, the qualitative result should have helped understand how the disentanglement happened.



**Summary Of The Paper:**

The paper is about learning disentanglement using scalable non-parametric density estimation. The author proposes a dual total correlation (DTC) metric for disentangling that divides the joint distribution into many low-dimensional conditional distributions. The authors conduct extensive experiments to show the superiority of their approach against state-of-the-art baselines.


**Summary Of The Review:**


Overall, the paper seems to be a work in progress, and there are many areas where it needs further improvement. Currently, the work has limited novelty and needs more experiments with results to justify the made claims.

---

> ### Author Response · Authors · 2022-11-19
> **Author response to questions (d23T)**
>
> Indeed, training discriminators can be a delicate process. However, we show in the appendix in the revision that training GCAE is faster than training FactorVAE with the recommended discriminator size on Beamsynthesis (5 discriminator iterations and $m=10$)
>
> This is a good point - we included more metrics in the revision (MIG, FactorScore, SAP, DCI Disentanglement). Together, they cover the 3 categories of metrics in "Measuring Disentanglement: A Review of Metrics". Thanks for pointing out that resource.
>
> The scatter plots in Figure 3 (a & b) are from both Beamsynthesis and dSprites - we observe similar trends from these two different data modalities. The relatively high absolute value of the correlation coefficient indicates a good predictive relationship between LSI and MIG. We do not differentiate between Beamsynthesis and dSprites in 3b using squares or circles - thanks for pointing this out. The data from both is there, though.
>
> We focused on reliability and performance of GCAE (and the baselines) on supervised metrics, so we did not consider qualitative comparisons in this work.
>
> Thank you for pointing out that we haven't adequately identified the role of lambda in 3a. We will be sure to clarify that in a future revision.
>
> Please see our response to atH7 regarding $\text{DTC}_i(z)$ - in short, it is equivalent to $\text{DTC}(z)$, but it is rewritten to emphasize that $z_i$ is only present as a conditioning variable (so gradient on DTC doesn't require knowledge of $p(z)$.

---

### Author Response · Authors · 2022-11-11
**Thank you for your constructive feedback**

We'd like to thank you all for your effort in providing these detailed reviews. The feedback from each reviewer is very helpful as we make improvements to this paper.

As requested, we are working on providing additional experiments (more datasets, recent baselines, and disentanglement metrics) to further support our claims. In the meantime, we will address the questions of each reviewer individually.

Thank you all again for your help.

---

### Author Response · Authors · 2022-11-19
**New Revision**

We made major improvements to the paper following the constructive feedback from all reviewers. Please see the new revision for these specific additions:

- Evaluation on four disentanglement metrics (MIG, FactorScore, SAP, and DCI Disentanglement). This group covers the 3 different metric categories identified by "Measuring Disentanglement: A Review of Metrics"
- Addition of the DIP-VAE-II baseline
- Comparison of GCAE with FactorVAE (closest TC-based method) when latent space dimension m is increased
- Detailed description of related work and metrics (in the appendix)
- Cleaned up mathematical notation throughout the paper
- Comparison of training time between GCAE (m discriminators) & FactorVAE (1 discriminator)

Thank you all for your time.

---

### Author Response · Authors · 2022-12-05
**Thank you for your review**

With one week left in the second discussion stage, we would be happy to answer any lingering questions regarding the revision.

The revision addresses many of the concerns from the reviewers. Some of the primary additions are:
 - Evaluation on 4 diverse disentanglement metrics (MIG, FactorScore, SAP, and DCI Disentanglement)
 - another baseline: DIP-VAE-II
 - Disentanglement performance as $m$ is changed

We would like to thank the reviewers for their time spent on this vital process.

---

### Decision · Program_Chairs · 2023-01-20

**Decision:**

Accept: poster

**Justification For Why Not Higher Score:**

The paper is a borderline accept and thus on the lower end of the score.

**Justification For Why Not Lower Score:**

Due to the significant additional experiments that the authors have performed during the rebuttal period the reviewers could be swayed to increase their scores wards an accept.

**Metareview: Summary, Strengths And Weaknesses:**

Strengths:
- better results on benchmark data sets than SOTA (strong baselines) 3
- proposed method circumvents curse of dimensionality
- well-motivated methods
- well written paper

Weakness:
- limited scope of comparison (metrics) 3 - added metrics in revision
- limited novelty in density estimation
- No qualitative results demonstrating disentanglement
- limited discussion of related work in density estimation (not addressed in rebuttal)

**Note From Pc:**

if the above contains the word "oral" or "spotlight" please see: "oral" presentation means -> notable-top-5% and "spotlight" means -> notable-top-25%. As stated in our emails, we are disassociating presentation type from AC recommendations

**Summary Of Ac-Reviewer Meeting:**

The reviewers agree that the main point raised by three out of four reviewers was the limited scope of comparison has been sufficeintly addressed in the rebuttal phase, where the authors added additional disentanglement metrics in their comparison.
While the main critique of the fourth reviewer, the limited discussion of related work on density estimation, had not been addressed, the reviewer still lean towards acceptance, as the application of the proposed density estimation approaches in disentanglement is sufficiently novel and interesting.